



# On the Climate Sensitivity and Historical Warming Evolution in Recent Coupled Model Ensembles

Clare Marie Flynn[1] and Thorsten Mauritsen[1]

[1]Department of Meteorology, Stockholm University, Stockholm, Sweden

**Correspondence:** C. M. Flynn (clare.flynn@misu.su.se)

**Abstract.** The Earth's equilibrium climate sensitivity (ECS) to a doubling of atmospheric $CO_2$, along with the transient climate response (TCR) and greenhouse gas emissions pathways, determines the amount of future warming. Coupled climate models have in the past been important tools to estimate and understand ECS. ECS estimated from Coupled Model Intercomparison Project Phase 5 (CMIP5) models lies between 2.0 and 4.7 K (mean of 3.2 K), whereas in the latest CMIP6 the spread has

increased: 1.8-5.5 K (mean of 3.7 K), with 5 out of 25 models exceeding 5 K. It is thus pertinent to understand the causes underlying this shift. Here we compare the CMIP5 and CMIP6 model ensembles, and find a systematic shift between CMIP eras to be unexplained as a process of random sampling from modeled forcing and feedback distributions. Instead, shortwave feedbacks shift towards more positive values, in particular over the Southern Ocean, driving the shift towards larger ECS values in many of the models. These results suggest that changes in model treatment of mixed-phase cloud processes and changes

to Antarctic sea ice representation are likely causes of the shift towards larger ECS. Somewhat surprisingly, CMIP6 models exhibit less historical warming than CMIP5 models; the evolution of the warming suggests, however, that several of the models apply too strong aerosol cooling resulting in too weak mid 20[th] Century warming compared to the instrumental record.

## 1   Introduction

The equilibrium climate sensitivity (ECS) is defined as the long term globally-averaged amount of surface temperature increase

in response to a doubling of atmospheric carbon dioxide ($CO_2$) relative to pre-industrial levels. An expression of ECS can be obtained from the linearised global radiation balance equation $N = F + \lambda T$, with $N$ the top-of-atmosphere (TOA) radiation balance, $F$ an external forcing, $\lambda$ the total feedback parameter and $T$ the global surface temperature change. Assuming a new equilibrium is reached ($N = 0$) after applying a sustained doubling of atmospheric $CO_2$ we obtain:

$$\text{ECS} = \frac{-F_{2x}}{\lambda}, \tag{1}$$

where $F_{2x}$ is the radiative forcing from a doubling of $CO_2$, equal to approximately 3.7 Wm$^{-2}$. Here $\lambda$ is the total climate feedback parameter in units of Wm$^{-2}$K$^{-1}$, defined as the sum over all feedback processes, including cloud, water vapor, lapse rate, surface albedo, Planck and other feedbacks. ECS endures as a key metric to examine the joint effect of forcing and feedback on the climate system, and for comparison of different climate models to each other (Andrews et al., 2012) and other lines of evidence besides climate models (Stevens et al., 2016).



Constraining the Earth's ECS is a critical problem in climate science, as an accurate estimate is necessary both for under-
standing the Earth's past climate changes, but also in practice to provide reliable projections of future warming (Collins et al.,
2013). Despite achieving equilibrium with the deep oceans requiring multiple millenia, Grose et al. (2018) found that ECS
explains more of the inter model spread in surface temperature change over the 21$^{st}$ Century than other metrics of climate
sensitivity, such as the commonly used transient climate response (TCR), which is the warming by the time of doubling in a

run with 1 percent increase in $CO_2$ per year. Unfortunately, the Intergovernmental Panel on Climate Change (IPCC) "likely"
range (greater than 66 percent probability) of 1.5-4.5 K for ECS, with a central estimate of about 3 K, has not significantly
changed since it was first proposed four decades ago by Charney et al. (1979) through to the Fifth IPCC Assessment Report
(AR5) (Collins et al., 2013).

    Early estimates of ECS were primarily based on various climate model results starting from the pioneering study of Arrhenius

(1896), though the IPCC AR5 report assessment includes other sources of evidence in addition to raw ECS estimates from
climate models. Recent community efforts to improve on this stalemate on bounding ECS instead focuses entirely on basic
process-understanding, historical warming and paleoclimate evidence (Stevens et al., 2016). This may be viewed as scientists
abandoning climate models as evidence for ECS, but this is not true. On the contrary models are used as tools in several places
within these three lines of evidence, e.g. to estimate forcing, parts of the feedback, and how temporary sea surface temperature

patterns might affect historical inference (Armour, 2017).

    In light of this, it is certainly valuable to understand how models obtain their respective ECS, and more so interesting that
the currently ongoing sixth phase of the Coupled Model Intercomparison Project (CMIP6) exhibit a marked increase in both
inter model mean (3.7 K) and range (1.8-5.5 K) in ECS, relative to the previous CMIP5 phase (3.2 K, 2.0-4.7). More so, the
CMIP6 models thus far exhibit an interesting bi-modal distribution (Fig. 1), indicative that systematic changes to some, but not

all models are responsible for the upward shift in model ensemble mean ECS.

    Indeed, recent studies of several individual CMIP6 models, including CNRM-CM6-1 (Voldoire et al., 2019), CESM2 (Gettel-
man et al., 2019), E3SMv1 (Golaz et al., 2019), and HadGEM3-GC3.1 (Bodas-Salcedo et al., 2019; Andrews et al., accepted),
each with an ECS of about 5 K or greater, have pointed to model parameterisation changes that increased the positive shortwave
cloud feedbacks or added aerosol-cloud interactions, as driving up their ECS values.

In this study we set out to investigate whether the collective shift in modelled ECS between CMIP5 and CMIP6 could have
happened by chance as the result of a random sampling process in model development, and whether the structure of the forcing
and feedback shows signs of systematic behavior across the ensembles. We round off by inspecting the ability of models to
represent the evolution of the instrumental record warming with a focus on early and late 20$^{th}$ Century warming. The results
allude to excessive aerosol cooling in early historical warming in a majority of the models.

## 2   Model Experiments and Methodology

The CMIP5 ensemble analyzed in this work includes 27 models, and the CMIP6 ensemble includes the 25 members available
at the time of writing. The first realization for each model (r1i1p1 for CMIP5 and r1i1p1f1 for CMIP6) was used, and all





climate model output was downloaded from the Earth System Federation Grid (ESGF) nodes. All models are listed in Tables 1 and 2 with their ECS, TCR, and feedback parameter values.

## 2.1 Estimation of Model Climate Sensitivities and Feedbacks

The ECS for each model was calculated from the CMIP *abrupt4xCO2* simulation, in which the $CO_2$ concentration is abruptly quadrupled at the beginning of the 150-year simulation and then held constant (Eyring et al., 2016). Since some models exhibit control state drift, accurate estimates of ECS and TCR require correcting for this, which we do here by assuming the underlying drift is approximately linear in time over the 150 years. The time slice of the pre-industrial control simulation (*piControl*), corresponding to the 150-year *abrupt4xCO2* simulation is first chosen, beginning at the simulation year at which *abrupt4xCO2* branched off of *piControl*. One must be cognisant that this information is not always reliable, so in a few cases the correction may not be accurate. A linear regression is then performed on the global annual mean *piControl* surface temperature or TOA radiative flux values to remove annual fluctuations, which is then used as the new *piControl*. The regression values are then subtracted from the global annual mean radiative fluxes and surface temperatures from *abrupt4xCO2* to obtain the radiative flux and surface temperature anomalies. These resulting anomalies are linearly regressed against each other, following the Gregory method (Gregory et al., 2004), to obtain the ECS value as one-half of the x-intercept and the total climate feedback parameter λ as the slope of the regression. Shortwave (SW) and longwave (LW) feedback parameters are calculated in a similar manner, but using the TOA SW radiative flux anomalies or LW radiative flux anomalies, respectively, instead of the total flux.

TCR is calculated from the *1pctCO2* CMIP simulation (Eyring et al., 2016), in which $CO_2$ is gradually increased at a rate of 1% per year. The corresponding time slice of *piControl* is first removed in the same manner as for ECS, to obtain the global annual mean *1pctCO2* surface temperature anomalies. TCR is then calculated as the mean surface temperature anomaly in a 20-year period centered on year 70 of the simulation; the year at which the $CO_2$ concentration is doubled.

## 2.2 Estimation of Model and Observational Historical Warming

Historical warming amounts were computed for each model. The early and late periods are defined as 1900-1969 (pre-1970s warming) and 1970-2005 (post-1970s warming), respectively, with years corresponding to the Agung and Pinatubo volcanic eruptions (1963-1964 and 1990-1993) excluded to limit the influence of natural volcanic aerosol forcing. Pre-1970s warming is strongly influenced by the uncertain aerosol cooling that off-set some of the greenhouse gas warming (Stevens, 2015), whereas post-1970s warming is dominated by greenhouse gas warming, while aerosol cooling only changed slightly, and so is expressive of TCR and ECS (Jiménez-de-la Cuesta and Mauritsen, 2019). The warming within each period is defined as the difference in the mean surface temperature between 1994-2005 and 1970-1989 for the late period, and between 1900-1939 and 1940-1969 for the early period.

Model historical warmings are compared to the same periods from the Cowtan and Way (2014) version 2.0 surface temperature reconstruction for years 1850 to present. In this reconstruction the land surface temperatures and sea surface temperatures (SST) are based on the HadCRUT version 4.2.0 and UAH version 5.6 global surface temperature datasets. Missing data are infilled by kriging. Data coverage uncertainty and ensemble uncertainty, or uncertainty arising from the choice of parameter





values used to create the reconstruction, are included in the data set. Uncertainty from natural variability within each warming period is computed based on the 100-member Max Planck Institut MPI-ESM1.1 model Grand Ensemble of historical climate change simulations (Maher et al., 2019), which is larger than the reconstruction uncertainties. Thus the total observational warming uncertainty is the sum in quadrature of coverage uncertainty, reconstruction parameter uncertainty, and uncertainty
due to natural climate variability.

## 3  Comparison of the Model Ensembles

In this section we shall first inspect the global ECS and feedback parameters in the two CMIP ensembles, and then we ask whether the shift could have happened by chance.

### 3.1  Shifts in Climate Sensitivity and Global Feedbacks between CMIP5 and CMIP6

Figure 1 displays the distributions of ECS for CMIP5 and CMIP6, with the mean value and standard deviation for each ensemble also displayed. The ensemble mean ECS increased from 3.2 K (range of 2.0-4.7 K) for CMIP5 to 3.7 K (1.8-5.5 K) for CMIP6, an increase of 0.5 K or 17%. Moreover, the CMIP6 distribution is shifted towards higher ECS, with a secondary peak at approximately 5 K. About 11% of CMIP5 models have an ECS greater than 4 K, compared to 40% of CMIP6 models. Only one CMIP6 model, INM-CM4-8, exhibits a relatively smaller ECS (1.81 K) than found in any model in CMIP5.

The average radiative forcing from $CO_2$, as estimated using the Gregory method (Gregory et al., 2004), does not change substantially between the CMIP ensembles, whereas the range narrows slightly (Fig. 2, Tables 1 and 2). The total feedback parameter $\lambda$ however, does exhibit an increase in ensemble mean, from -1.13 Wm$^{-2}$K$^{-1}$ ($\pm$ 0.28) to -1.02 Wm$^{-2}$K$^{-1}$ ($\pm$ 0.32). This shift towards less negative values is also discernible in Fig. 2, particularly for models with ECS on the high end. Therefore, the decrease in $\lambda$ magnitudes, which alone determines most of the variation in ECS, is the main driver behind the shift toward
higher ECS between the CMIP ensembles.

### 3.2  Could We Obtain the CMIP6 Ensemble Mean ECS by Chance?

The results presented in Section 3.1 demonstrated a clear shift in ECS. One might view such ensembles as small random samples taken from some generic modelling activities. In this view, how likely is it that we obtain the CMIP6 ensemble mean ECS increase simply by chance? In other words, do the high CMIP6 climate sensitivities represent a statistically significant
shift in an envisioned underlying probability distribution based on modeling, or are they encapsulated by the uncertainty of climate modeling? We address this question by assuming the underlying distribution is well described by Equation (1).

First, one must understand that the mean of the resulting distribution is generally larger than the median (Roe and Baker, 2007); it should be noted that using the mean $\lambda$ and $F_{2x}$ in Equation (1) represents the median ECS of the underlying distribution. To show this, we Monte-Carlo sample feedback parameters from Gaussian distributions with a standard deviation equal
to the average of the CMIP5 and CMIP6 ensemble standard deviations (0.29 Wm$^{-2}$K$^{-1}$, Tables 1 and 2), and forcing centered on 3.7 Wm$^{-2}$ with a standard deviation of 10 percent. Negative ECS as well as values exceeding 10,000 K are omitted. For each





value of median ECS we can then evaluate the resulting mean, which is quite close for lower values of ECS but diverges for higher sensitivities (Fig. 3, upper panel). For the CMIP5 mean of 3.2 K, the corresponding median is 3.0 K, and for the CMIP6 mean of 3.7 K the median is 3.4 K.

Using these medians we next address the question of whether CMIP6 could be obtained simply by chance. To do so, we first assume the underlying median ECS is 3.0 K and make 100,000 random ensembles each with 25 models. The resulting distribution is shown in Figure 3. It turns out that less than 2 percent of the samples exceed 3.7 K, which is the mean of CMIP6. Likewise, if we assume the underlying median is 3.4 K, centered at CMIP6, then less than 2 percent of the samples have a mean less than 3.2 K, which is the mean of CMIP5. Thus, even though the change in ensemble mean feedback parameter is

fairly small, which in turn caused the shift in ECS, it is extremely unlikely to have been caused simply by chance.

## 4    Decomposition into Longwave and Shortwave Feedbacks

Having established that there is a systematic shift in feedback underlying the increase in ensemble mean ECS from CMIP5 to CMIP6, we next divide the feedback into longwave, shortwave, all-sky and clear-sky components, and inspect the zonal mean distribution in order to seek the possible underlying causes.

### 4.1    Global-Mean All-Sky and Clear-Sky Feedbacks

Decomposition of the total feedback parameter into the all-sky shortwave (SW; $\lambda_{SW}$) and longwave (LW; $\lambda_{LW}$) components, and examination of the clear-sky (CS) SW and LW feedbacks ($\lambda_{CS,SW}$; $\lambda_{CS,LW}$), elucidates which classes of feedbacks drive the increase in ECS. As shown in Fig. 4 (top panel), a systematic shift toward more positive $\lambda_{SW}$ has occurred on average for the CMIP6 ensemble relative to CMIP5: the mean $\lambda_{SW}$ increased from 0.64 to 0.81 $Wm^{-2}K^{-1}$, whereas the mean $\lambda_{LW}$ remained

almost unchanged (mean of -1.74 and -1.78 $Wm^{-2}K^{-1}$, respectively). However, much spread in the SW and LW feedbacks exists within both ensembles as indicated by the large standard deviations.

The shortwave feedback parameters are strongly associated with the total feedback parameter for both model ensembles, with a correlation coefficient of 0.83 (p-value less than 0.001) for CMIP5 and 0.56 (p-value of 0.004) for CMIP6, whereas the longwave feedbacks exhibited small, statistically non-significant correlations with the total feedback parameter (-0.21 and 0.11

for CMIP5 and CMIP6, respectively). The longwave thus exhibits no consistent or systematic shift with ECS, whereas these results suggest that $\lambda_{SW}$ is the main cause of both the variations and the shift in $\lambda$ and thus of ECS. These feedbacks suggest that much of the spread is caused by cloud parameterisations, and that cloud feedbacks play an important role in the shift to higher ECS in CMIP6.

In contrast, no systematic shifts are evident in the clear-sky feedback parameters ($\lambda_{CS,SW}$ or $\lambda_{CS,LW}$) between the CMIP

eras (Fig. 4), and again much spread among models is evident in both ensembles. However, the spread in CMIP6 $\lambda_{CS,SW}$ is smaller than that for CMIP5, with a standard deviation of 0.13 compared to 0.18 $Wm^{-2}K^{-1}$, indicating a greater convergence of the CMIP6 $\lambda_{CS,SW}$ values, while the standard deviations for the clear-sky longwave feedbacks are of similar magnitude (0.12 $Wm^{-2}K^{-1}$). This is in contrast to the all-sky feedbacks, where the standard deviations were larger for both SW and LW for





CMIP6. Lastly, the clear-sky feedbacks in Fig. 4 (bottom panel) do not exhibit a statistically significant slope for both ensembles

despite the spread among models, whereas the all-sky feedbacks (Fig. 4, top panel) exhibited statistically significant, negative slopes (-0.37 and -0.47 for CMIP5 and CMIP6, respectively); the dominant direction of the spread has changed between all-sky and clear-sky. Thus, another feedback besides cloud feedback may be be causing the spread, such as the surface albedo feedback, and it is notable that the spread in $\lambda_{CS,SW}$ decreased between CMIP5 and CMIP6.

## 4.2  Zonal-Mean Feedbacks

The all-sky and clear-sky feedback parameters are decomposed into zonal-mean feedback parameters, to further investigate the the causes of the shifts in the shortwave feedbacks and which regions may be the main drivers. The zonal-mean feedbacks are calculated similarly to the global-mean, annual-mean feedbacks, with the exception that the global-mean, annual-mean surface temperature anomalies are regressed instead against zonal-mean, annual-mean TOA imbalances. The radiation fluxes are first divided into 10° latitude bins based on each model's grid, centered between 85°S to 85°N, and then the Gregory method is

applied to compute the zonal-mean all-sky and clear-sky feedbacks. These feedbacks as a function of latitude are displayed in Fig. 5 for all-sky and Fig. 6 for clear-sky.

Large differences in all-sky feedbacks between CMIP eras tend to occur in the Tropics and towards the poles. In particular, a broad swath of change is seen for the Southern Hemisphere mid-latitude and polar regions; the largest shortwave feedback differences are found in these regions, where the CMIP6 zonal shortwave feedbacks have substantially increased (Fig. 5).

Though smaller in magnitude, clear-sky zonal shortwave feedback also shows substantial increases between CMIP5 and CMIP6 poleward of 55°S, in the Southern Ocean (Fig. 6). The broad increases from CMIP5 to CMIP6 in all-sky $\lambda_{SW}$ and $\lambda_{LW}$ across much of the Southern Hemisphere extratropics, coupled with changes in clear-sky feedback only within the Southern polar regions, indicate that cloud feedbacks have changed between CMIP eras. It is also notable that the variability among models within the CMIP6 ensemble has decreased relative to CMIP5 in the shortwave for both all-sky and clear-sky, as indicated by

the smaller standard deviation bounds on the ensemble averages in Figs. 5 and 6.

The largest changes for the clear-sky feedbacks occur for $\lambda_{CS,SW}$ over the Southern Ocean latitudes (Fig. 6, middle and bottom panels), where a shift towards more positive clear-sky shortwave feedback is found. This is suggestive of increases in the sea ice induced surface albedo feedback, likely due to increased abundance of sea ice near the Antarctic in the underlying *piControl* climatology in CMIP6 relative to CMIP5 (Fig. 7). Perhaps as a result of this larger base-state abundance, the decrease

in sea ice coverage in the Antarctic in the *abrupt4xCO2* is also greater for CMIP6 than CMIP5 (Fig. 7). Larger decreases in sea ice coverage for *abrupt4xCO2* are also seen in the Arctic, but accompanied by a much smaller change in shortwave feedback relative to the Antarctic. Furthermore, in contrast to the Antarctic, the difference in net feedback in the Arctic is smaller than for the Antarctic, and the change in the clear-sky SW feedback in Northern Hemisphere mid-latitudes is negative (Fig. 6). Perhaps as a result there is less intense Arctic amplification exhibited by CMIP6 relative to CMIP5 (Fig. 8). Surface temperature

increases in the Arctic still exceeds warming elsewhere in the CMIP6 ensemble, but of a somewhat smaller magnitude than CMIP5, due to a relatively lessened impact of sea ice albedo on the feedback parameter.





We speculate that much of this behaviour can be explained by an increased focus on the representation of mixed-phase clouds by the models micro-physics parameterisations. Recent studies have shown that the strength of the negative cloud optical depth feedback is strongly dependent on the relative partitioning of ice and liquid phase cloud condensate in the control state (Tan
et al., 2016). By increasing the amount of liquid in super-cooled clouds the negative optical depth feedback is weakened and hence ECS increases. In addition, since liquid clouds are generally more reflective than ice clouds the long-standing Southern Ocean warm bias may have been reduced through these efforts, thereby resulting in more abundant sea ice. These effects could, together, explain the non-trivial increase in ECS in the CMIP6 ensemble over CMIP5.

## 5  Transient Climate Response, Historical Warming and Aerosol Cooling

The instrumental record warming is the prima facie test of climate models: if models are not able to reproduce the history of warming then they do not represent a credible hypothesis of how the climate system works. However, the warming in a model is a result of both climate change feedbacks, radiative forcing, deep ocean heat uptake and pattern effects and therefore modellers can trade off these factors to obtain an overall warming in line with observations (Kiehl, 2007). Some modelling centres use this explicitly to tune their models (Hourdin et al., 2017; Mauritsen et al., 2019) whereas others state they do not
do this (Schmidt et al., 2017). In either case, as such representing historical warming is a necessary, but insufficient validation of a climate model.

A central metric that incorporates several of the factors relevant for historical warming is the transient climate response (TCR). TCR is computed from an idealized simulation with a gradual 1% per year $CO_2$ increase as the warming around the time of doubling. Just as ECS, also TCR has increased in CMIP6 to a mean of 1.98 K (range 1.30-2.91 K) compared to the
CMIP5 mean of 1.75 K (0.96-2.58 K), as seen in Fig. 9. One can obtain an approximate estimate of TCR in terms of physical bulk properties of the climate system (Jiménez-de-la Cuesta and Mauritsen, 2019):

$$\text{TCR} \approx \frac{-F_{2x}}{\lambda - \epsilon\gamma}, \tag{2}$$

where the product $\epsilon\gamma$ is equal to 0.93 Wm$^{-2}$K$^{-1}$ with an uncertainty range of 0.54-1.32 Wm$^{-2}$K$^{-1}$ in CMIP5 (Geoffroy et al., 2013); $\epsilon$ is the deep ocean heat uptake efficacy representative of forced temporary pattern effects, and $\gamma$ is the deep ocean heat
uptake coefficient. The product $\epsilon\gamma$ controls the relationship between TCR and ECS. Models in CMIP6 follow the predicted behavior of Equation (2) using CMIP5 parameters surprisingly well (Fig. 9).

Given that TCR is on average higher in CMIP6 one might naively expect stronger historical warming; however, this is not the case (Fig. 10). Whereas CMIP5 on average tracked the instrumental record quite well, warming slightly too much in the latter half of the 20[th] Century, the CMIP6 models are systematically on average colder than observed starting around 1940, but
nearly catching up with global warming in the beginning of the 21[st] Century. Looking at individual model simulations (Fig. 11) reveals that also the spread in overall centennial warming increased in CMIP6, and furthermore that there is not a strong relationship with TCR.





To demonstrate at this point that the most likely explanation for why CMIP6 on average warms less is because of stronger aerosol cooling we divide warming into the pre-1970s and post-1970s (Fig. 12). The rationale behind this division is that aerosol cooling, which has off-set some of the greenhouse gas warming, increased rapidly with industrialisation up until around 1970, where after air quality regulations have resulted in stabilised global aerosol cooling. Since the amount of anthropogenic aerosol cooling, in contrast to greenhouse gas warming, is highly uncertain (Bellouin et al.) and varies among models, total forcing uncertainty in the pre-1970s period is dominating the global temperature response (Stevens, 2015). In the post-1970s period instead, the greenhouse gas forcing change dominates and is less uncertain, such that the variations in TCR are more important (Jiménez-de-la Cuesta and Mauritsen, 2019).

Interestingly, the majority of models from both ensembles underpredicts the pre-1970s warming (Fig. 12), with several CMIP6 models exhibiting close to no warming. This is a strong indication that many models apply too strong aerosol cooling, and that this is more outspoken in CMIP6. More than half the models, however, make up for this lack of warming by instead warming more than observed in the post-1970s period. As expected, there is no apparent relationship between pre-1970s warming and TCR, but a correlation exists with post-1970s warming, more apparent for models with TCR of 1.5-2.0 K. None of the models with TCR greater than 2.5 K provide a realistic post-1970s warming.

## 6 Conclusions

We have compared the CMIP5 and CMIP6 model ensembles in terms of their climate sensitivities, feedback parameters, and historical warming evolution. The ECS and total feedback parameter values were computed with the Gregory method, and we found that both the ensemble mean ECS and the spread in ECS values has increased between CMIP5 (mean 3.2 K, spread 2.0-4.7 K) and CMIP6 (mean 3.7 K, spread 1.8-5.5 K).

We examined whether this shift in ECS between ensembles could have arisen simply by chance, or whether it is a statistically significant change. This is a critical question, because it speaks to whether such a shift in ECS is truly unexpected or not. We modeled distributions of forcing and feedbacks as random samples from Gaussian distributions centered at CMIP5, and determined that the probability of obtaining the CMIP6 ensemble mean ECS value was less than 2 percent. Previous model ensemble mean ECS values are similar to that obtained for the CMIP5 ensemble, together suggesting that the CMIP6 ensemble mean ECS is indeed highly unusual.

This shift towards higher ECS for the CMIP6 ensemble is primarily driven by increases in the shortwave feedback parameter for some models within the ensemble. The mean total feedback parameter increased from -1.13 $Wm^{-2}K^{-1}$ for CMIP5 to -1.02 $Wm^{-2}K^{-1}$ for CMIP6, and the mean all-sky shortwave feedback parameter increased from 0.64 $Wm^{-2}K^{-1}$ to 0.81 $Wm^{-2}K^{-1}$. While the all-sky shortwave feedback parameters exhibited statistically significant correlations with the total feedbacks for each CMIP ensemble, no statistically significant correlation or systematic change was seen for the longwave feedback parameters. This constitutes a systematic shift in feedbacks underlying the increase in ensemble mean ECS, and are suggestive of the role of cloud feedback processes. The global and zonal clear-sky shortwave feedback parameters also suggested a significant role for the albedo feedback in the increase in ECS, likely driven by increases in Southern Ocean sea ice coverage in CMIP6 relative



to CMIP5. We speculate that these results are due to changes in model treatment of mixed-phase cloud processes reducing the negative optical depth cloud feedback, and resulting changes to Antarctic sea ice representation, and are the likely cause of the systematic shift towards larger ECS.

Lastly, we examined the historical warming in the model ensembles, which surprisingly despite an increase in ECS and TCR is weaker in CMIP6 than in CMIP5. Whereas CMIP5 models on average track the instrumental record warming fairly well, CMIP6 models are colder than observed from around 1940 and onwards, to only catch up with global warming in the early $21^{st}$ Century. Detailed examination of pre- and post-1970s warming is suggestive that the majority of climate models from both ensembles exaggerate anthropogenic aerosol cooling, but that this is more so the case for some CMIP6 models. Models that best agree with observations of post-1970s warming tend to have mid-range TCR, whereas no model with a TCR above 2.5 K
matches observations.

*Author contributions.* The analysis of CMIP models was performed by CMF. TM provided the initial project idea. The manuscript was written by CMF and TM.

*Competing interests.* The authors declare that they have no conflict of interest.

*Acknowledgements.* We thank Diego Jiménez de la Cuesta, Martin Renoult, Navjit Sagoo, Kari Alterskjær, and Mark Zelinka for useful
comments and discussions that helped advance this study. This work was supported by the Stockholm University Faculty of Science, and through funding from the European Research Council (ERC) Grant agreement 770765 and the European Union's Horizon 2020 program Grant agreement 820829. We acknowledge the World Climate Research Programme's (WCRP) Working Group on Coupled Modelling (WGCM) and the climate modeling groups listed in Tables 1 and 2 for producing and making their output available.



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





**Table 1.** List of CMIP5 Models and model climate parameters.

| Model | ECS | TCR | $F_{2x}$ | $\lambda$ | $\lambda_{SW}$ | $\lambda_{LW}$ | $\lambda_{CS,SW}$ | $\lambda_{CS,LW}$ |
|---|---|---|---|---|---|---|---|---|
| MPI-ESM-LR | 3.48 | 1.94 | 4.05 | -1.16 | 0.51 | -1.68 | 0.73 | -1.85 |
| MPI-ESM-MR | 3.31 | 1.93 | 4.03 | -1.22 | 0.57 | -1.78 | 0.69 | -1.91 |
| MPI-ESM-P | 3.31 | 1.96 | 4.24 | -1.28 | 0.42 | -1.71 | 0.68 | -1.86 |
| MIROC5 | 2.70 | 1.49 | 4.09 | -1.51 | 0.39 | -1.90 | 0.85 | -1.86 |
| MIROC-ESM | 4.68 | 2.15 | 4.23 | -0.90 | 0.99 | -1.89 | 0.82 | -1.91 |
| IPSL-CM5B-LR | 2.58 | 1.44 | 2.64 | -1.02 | 0.89 | -1.91 | 0.57 | -1.89 |
| IPSL-CM5A-MR | 4.03 | 1.96 | 3.30 | -0.82 | 1.05 | -1.87 | 0.44 | -2.01 |
| IPSL-CM5A-LR | 3.97 | 1.94 | 3.17 | -0.80 | 1.17 | -1.97 | 0.46 | -2.01 |
| INM-CM4 | 2.01 | 1.22 | 2.91 | -1.45 | 0.59 | -2.04 | 0.68 | -2.01 |
| CSIRO-Mk3.6.0 | 4.05 | 1.76 | 2.58 | -0.64 | 1.32 | -1.96 | 0.85 | -1.71 |
| CNRM-CM5 | 3.21 | 2.04 | 3.67 | -1.14 | 0.49 | -1.63 | 0.79 | -1.73 |
| CNRM-CM5-2 | 3.40 | 1.63 | 3.68 | -1.08 | 0.58 | -1.66 | 0.90 | -1.73 |
| BNU | 3.98 | 2.58 | 3.71 | -0.93 | 0.66 | -1.60 | 1.10 | -1.75 |
| BCC-CSM1.1 | 2.81 | 1.74 | 3.36 | -1.19 | 0.49 | -1.69 | 0.77 | -1.90 |
| BCC-CSM1.1(m) | 2.77 | 2.00 | 3.88 | -1.40 | 0.50 | -1.90 | 0.49 | -1.97 |
| MRI-CGCM3 | 2.65 | 1.58 | 3.20 | -1.21 | 0.92 | -2.13 | 0.81 | -1.93 |
| NORESM1-M | 2.75 | 1.34 | 3.05 | -1.11 | 0.70 | -1.82 | 0.86 | -1.87 |
| ACCESS1.0 | 3.76 | 1.72 | 2.87 | -0.76 | 0.78 | -1.54 | 0.76 | -1.62 |
| CanESM2 | 3.71 | 2.37 | 3.72 | -1.00 | 0.40 | -1.40 | 0.74 | -1.86 |
| GFDL-ESM2M | 2.33 | 1.23 | 3.27 | -1.40 | 0.62 | -1.68 | 0.61 | -1.69 |
| GFDL-ESM2G | 2.30 | 0.96 | 3.00 | -1.30 | 0.55 | -1.59 | 0.64 | -1.70 |
| GFDL-CM3 | 3.85 | 1.85 | 2.95 | -0.77 | 1.27 | -2.03 | 0.71 | -1.97 |
| CCSM4 | 2.90 | 1.64 | 3.43 | -1.18 | 0.68 | -1.86 | 0.94 | -1.94 |
| FGOALS-g2 | 3.39 | 1.42 | 2.79 | -0.82 | 0.76 | -1.54 | 1.01 | -1.71 |
| GISS-E2-H | 2.33 | 1.69 | 3.72 | -1.60 | -0.22 | -1.37 | 0.54 | -1.65 |
| GISS-E2-R | 2.06 | 1.41 | 3.66 | -1.78 | -0.37 | -1.44 | 0.41 | -1.96 |
| HADGEM2-ES | 3.96 | 2.38 | 3.63 | -0.92 | 0.63 | -1.54 | 0.42 | -1.68 |
| Ensemble Mean ± Std: | 3.20 ± 0.70 | 1.75 ± 0.38 | 3.44 ± 0.48 | -1.13 ± 0.28 | 0.64 ± 0.37 | -1.75 ± 0.37 | 0.71 ± 0.18 | -1.84 ± 0.12 |





**Table 2.** List of CMIP6 Models and model climate parameters.

| Model | ECS | TCR | $F_{2x}$ | $\lambda$ | $\lambda_{SW}$ | $\lambda_{LW}$ | $\lambda_{CS,SW}$ | $\lambda_{CS,LW}$ |
|---|---|---|---|---|---|---|---|---|
| MIROC6 | 2.60 | 1.58 | 3.61 | -1.39 | 0.61 | -2.05 | 0.83 | -1.98 |
| IPSL-CM6A-LR | 4.50 | 2.39 | 3.39 | -0.75 | 1.10 | -1.66 | 0.62 | -1.51 |
| CNRM-CM6-1 | 4.81 | 2.23 | 3.70 | -0.77 | 0.68 | -1.45 | 0.77 | -1.76 |
| BCC-CSM2-MR | 3.07 | 1.60 | 3.06 | -1.00 | 0.79 | -1.79 | 0.71 | -1.91 |
| MRI-ESM2 | 3.11 | 1.67 | 3.37 | -1.08 | 0.84 | -1.93 | 0.84 | -1.95 |
| CanESM5 | 5.58 | 2.75 | 3.68 | -0.66 | 0.70 | -1.36 | 0.78 | -1.86 |
| CESM2 | 5.15 | 1.99 | 3.19 | -0.62 | 1.32 | -1.94 | 0.54 | -1.80 |
| GISS-E2-1-H | 2.99 | 1.81 | 3.47 | -1.16 | 0.21 | -1.37 | – | – |
| GISS-E2-1-G | 2.60 | 1.66 | 3.84 | -1.48 | -0.04 | -1.44 | – | – |
| SAM0-UNICON | 3.30 | 2.08 | 4.49 | -1.36 | 1.29 | -2.65 | 0.82 | -2.01 |
| E3SM-1-0 | 5.09 | 2.91 | 3.39 | -0.67 | 1.21 | -1.88 | 0.54 | -1.78 |
| UKESM1-0-LL | 5.31 | 2.79 | 3.56 | -0.67 | 1.59 | -2.26 | 0.72 | -1.91 |
| CNRM-ESM2-1 | 4.75 | 1.82 | 2.96 | -0.62 | 0.72 | -1.35 | 0.75 | -1.59 |
| BCC-ESM1 | 3.29 | 1.77 | 3.02 | -0.92 | 0.65 | -1.57 | 0.69 | -1.83 |
| CESM2-WACCM | 4.65 | 1.92 | 3.26 | -0.70 | 1.34 | -2.04 | 0.31 | -1.86 |
| MIROC-ES2L | 2.66 | 1.51 | 4.03 | -1.51 | 0.38 | -1.89 | 0.76 | -1.87 |
| EC-EARTH3-VEG | 3.93 | 2.76 | 3.53 | -0.90 | 0.78 | -1.69 | 0.86 | -1.63 |
| HADGEM3-GC31-LL | 5.46 | 2.47 | 3.48 | -0.64 | 1.64 | -2.28 | 0.67 | -1.83 |
| NORCPM-1 | 2.78 | 1.55 | 3.58 | -1.29 | 0.62 | -1.89 | 0.82 | -1.90 |
| GFDL-CM4 | 3.79 | – | 3.14 | -0.83 | 0.77 | -1.59 | 0.80 | -1.79 |
| GFDL-ESM4 | 2.56 | – | 3.84 | -1.50 | 0.13 | -1.63 | – | – |
| NESM3 | 4.50 | – | 3.78 | -0.84 | 0.61 | -1.45 | 0.81 | -1.69 |
| NORESM2-LM | 2.49 | 1.48 | 3.44 | -1.38 | 1.46 | -1.89 | 0.57 | -1.75 |
| MPI-ESM1-2-HR | 2.84 | 1.57 | 3.60 | -1.27 | 0.22 | -1.49 | 0.63 | -1.90 |
| INM-CM4-8 | 1.81 | 1.30 | 2.64 | -1.46 | 0.53 | -1.99 | 0.79 | -1.88 |
| Ensemble Mean ± Std: | 3.74 ± 1.11 | 1.98 ± 0.48 | 3.48 ± 0.37 | -1.02 ± 0.32 | 0.81 ± 0.45 | -1.78 ± 0.45 | 0.71 ± 0.13 | -1.82 ± 0.12 |



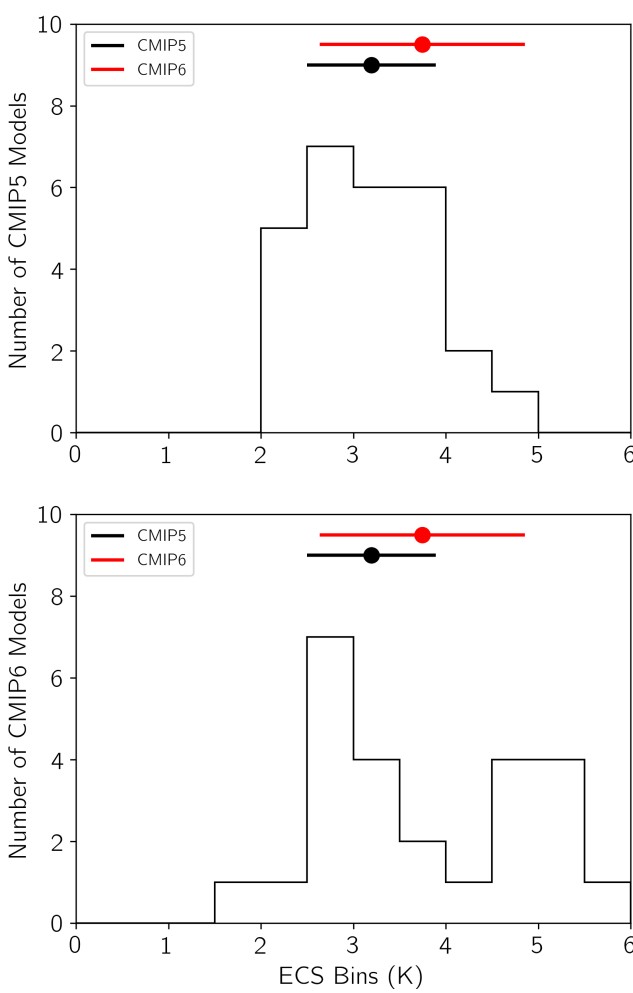

**Figure 1.** Histograms displaying number of CMIP5 (left) or CMIP6 (right) models that fall within 0.5 K ECS bins. ECS mean value and standard deviation for CMIP5 and CMIP6 ensemble displayed in black and red, respectively, above each histogram.

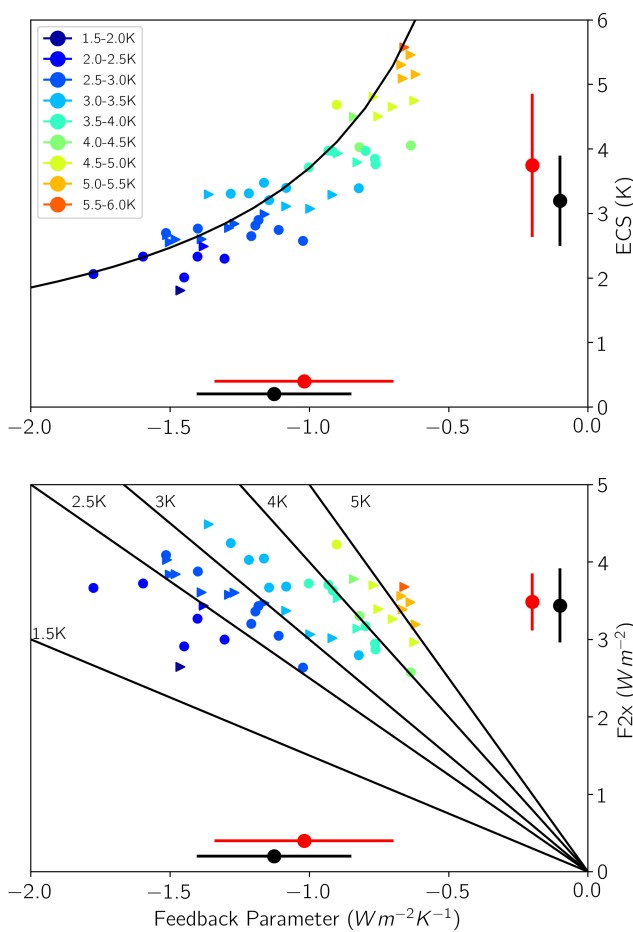

**Figure 2.** ECS vs. total net feedback parameter. Black curve represents the expected ECS value based on a forcing of 3.7 Wm$^{-2}$ over the range of feedbacks plotted (top), and effective forcing vs. total net feedback parameter. Black lines represent the expected forcing-feedback relationship based on the ECS value given in the label of each line (bottom). Circles represent CMIP5 models, right-facing triangles represent CMIP6 models. Mean value and standard deviation for each parameter for CMIP5 and CMIP6 ensemble displayed in black and red, respectively, on the appropriate axis in each plot. Plot symbols colored by ECS values as shown in legend.



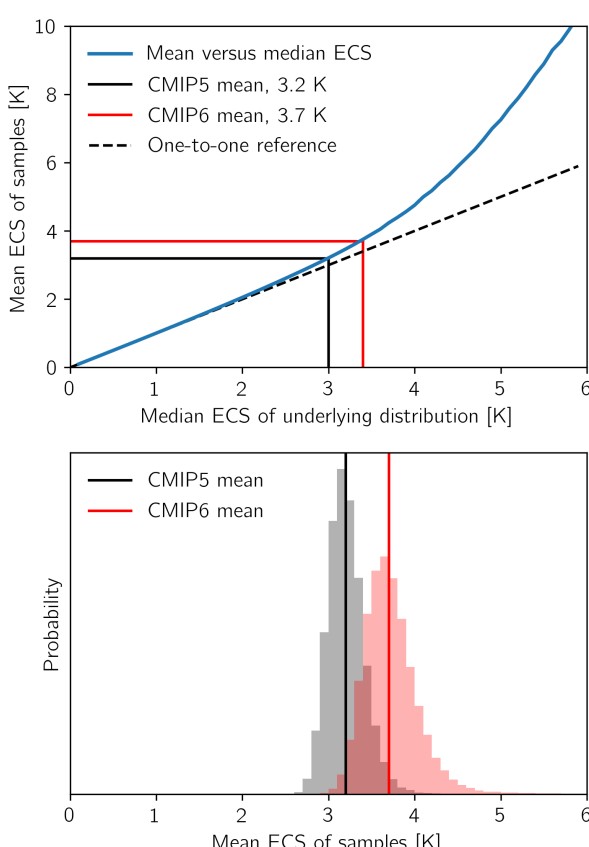

**Figure 3.** Random sampling of ECS from Gaussian distributions of $\lambda$ and $F_{2x}$. Upper panel shows relationship between the median and mean of ECS, arising from the inverse relationship between ECS and $\lambda$. Lower panel shows distributions of mean ECS from random 25 member ensembles centered at the means of CMIP5 and CMIP6.

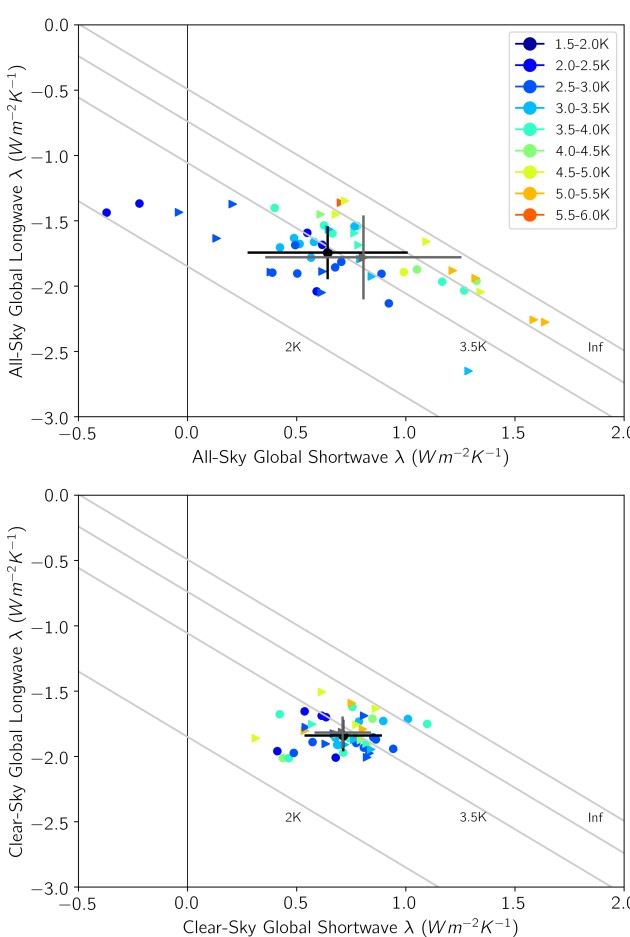

**Figure 4.** All-sky $\lambda_{LW}$ vs. $\lambda_{SW}$ for the CMIP5 and CMIP6 ensemble (top), and clear-sky $\lambda_{CS,LW}$ vs. $\lambda_{CS,SW}$ for CMIP5 and CMIP6 (bottom). CMIP5 as circles and CMIP6 as right-facing triangles. Mean CMIP5 feedbacks and standard deviations as black circle and lines, and mean CMIP6 and standard deviations as dark gray triangle and lines in each plot. Lines of constant ECS based on forcing of 3.7 Wm$^{-2}$ in light gray. Plot symbols colored by ECS values as shown in legend.



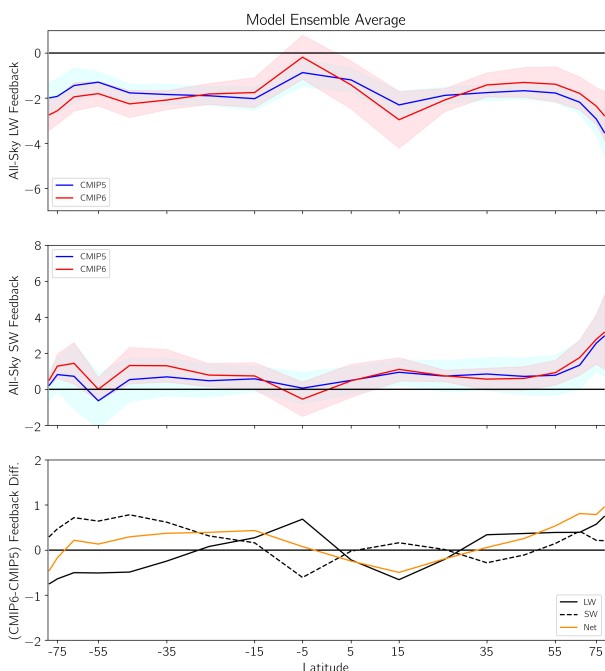

**Figure 5.** All-sky zonal average $\lambda_{LW}$ (top) and $\lambda_{SW}$ (middle) for the CMIP5 ensemble average (blue) and CMIP6 ensemble (red). Light blue and red shading represent standard deviation of each ensemble. Bottom panel displays the difference between the CMIP6 and CMIP5 ensemble average SW, LW, and net feedbacks as a function of latitude.



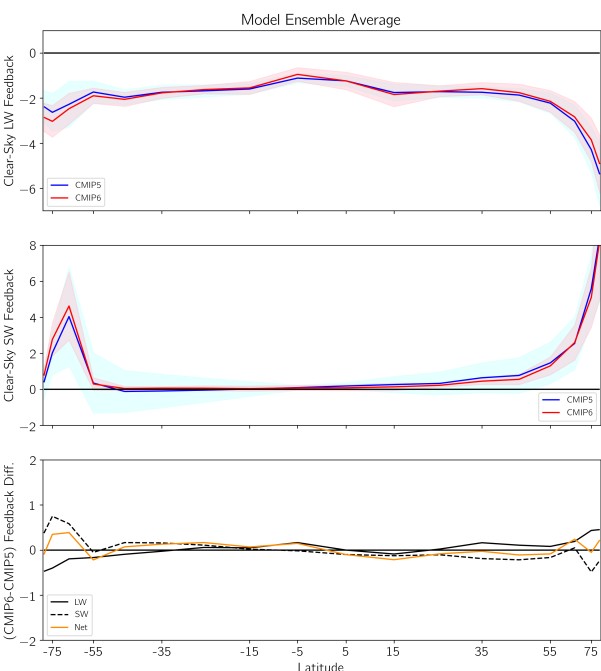

**Figure 6.** Clear-sky zonal average $\lambda_{CS,LW}$ (top) and $\lambda_{CS,SW}$ (middle) for the CMIP5 ensemble average (blue) and CMIP6 ensemble (red). Light blue and red shading represent standard deviation of each ensemble. Bottom panel displays the difference between the CMIP6 and CMIP5 ensemble average SW, LW, and net feedbacks.





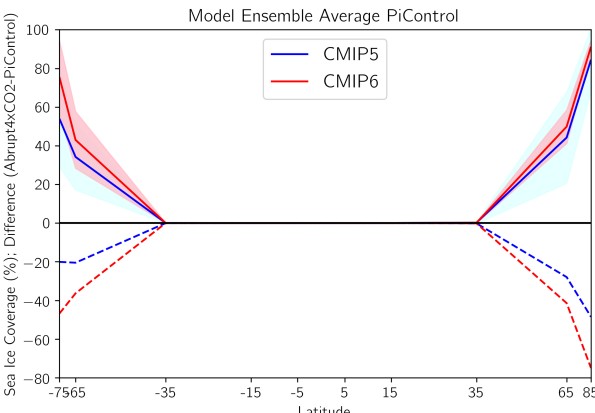

**Figure 7.** Zonal average sea ice coverage for the CMIP5 ensemble average (blue) and CMIP6 ensemble (red). Light blue and red shading represent standard deviation of each ensemble. Dashed lines represent the zonal average difference between the *piControl* simulation and the mean of the last 30 years of the *abrupt4xCO2* simulation.



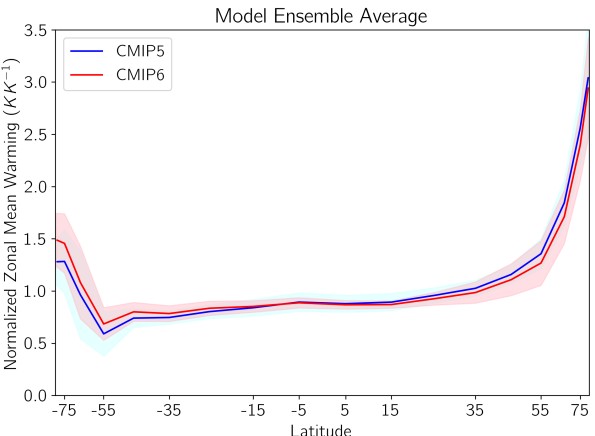

**Figure 8.** Zonal average surface temperature anomaly for the CMIP5 ensemble average (blue) and CMIP6 ensemble (red). Light blue and red shading represent standard deviation of each ensemble.



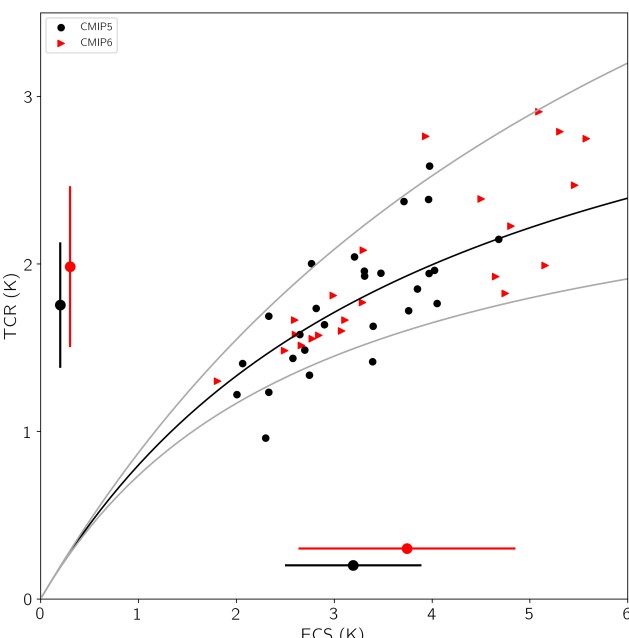

**Figure 9.** TCR vs. ECS for the CMIP5 ensemble (black, circles) and CMIP6 ensemble (red, right-facing triangles). Expected values based on forcing of 3.7 Wm$^{-2}$ and value of $\epsilon\gamma = 0.93$ Wm$^{-2}$K$^{-1}$ is the black curve, and uncertainty of $\epsilon\gamma$ value as gray bounding lines. ECS and TCR mean values and standard deviations for CMIP5 and CMIP6 ensemble displayed in black and red, respectively.

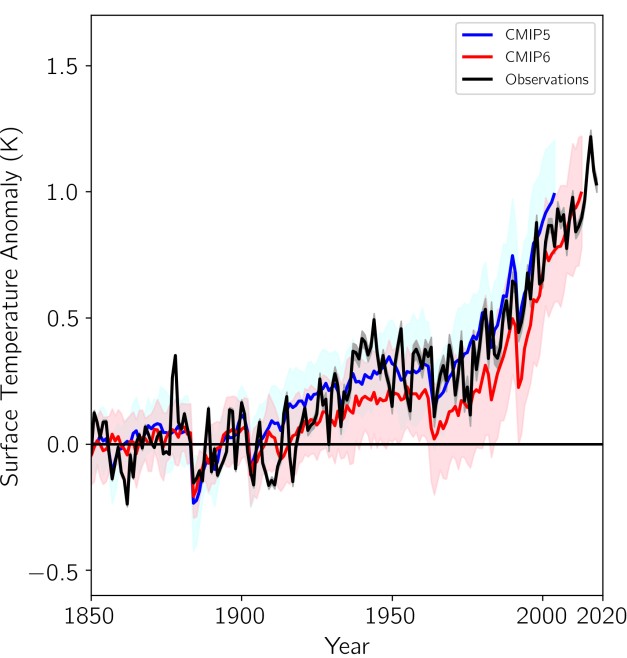

**Figure 10.** Ensemble mean historical surface warming in CMIP5 and CMIP6 compared with observations. Shading on the models is the ensemble standard deviation. The baseline is 1850-1900.

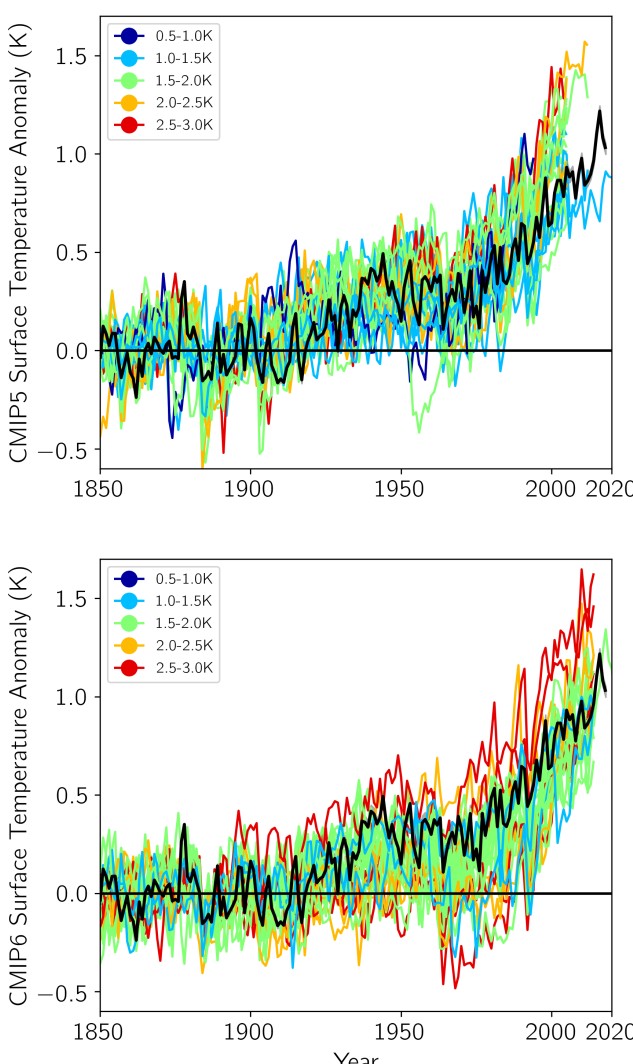

**Figure 11.** As in Figure 10 but for individual model runs. Upper panel shows CMIP5 models and lower panel shows CMIP6 models. Color coding is according to the respective models' TCR.

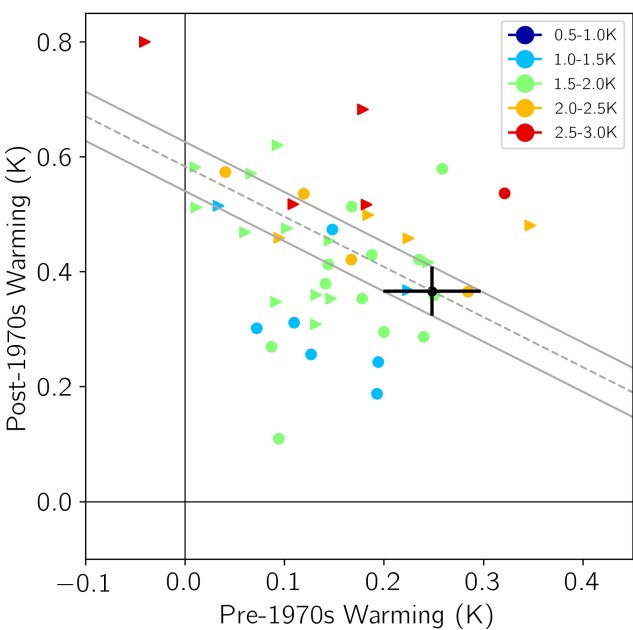

**Figure 12.** Post-1970s warming (surface temperature change between (1970-1990) and (1994-2005) periods) vs. pre-1970s warming (surface temperature change between (1900-1939) and (1940-1969) periods), with plot symbols colored by TCR bins shown in the legend. Circles represent CMIP5 models, right-facing triangles represent CMIP6 models. Observational pre- and post-1970s warmings plotted as black circle with uncertainty as black lines. Solid gray lines represent outer bounds of pre- and post-1970s warming summing to total observed warming.