# Peer review of "On the Climate Sensitivity and Historical Warming Evolution in Recent Coupled Model Ensembles"

_Atmospheric Chemistry and Physics, 2019_

## Referee Comment (RC1) · Anonymous Referee #1 · 17 Feb 2020

This paper is an analysis of the differences between CMIP5 and CMIP6. It has relatively low information content, but I don't see anything technically wrong with it. My comments are therefore relatively minor.

Overall, having read the paper, it was hard to determine what the top-line scientific message of the paper was. This problem is epitomized in the abstract. Per that, the most important conclusion is that the difference in ensemble averages between CMIP5 and CMIP6 could not have arisen by chance. Is that really the most important point in the paper? I do think it's a reasonable point to make, but it's a simple means test and it's hard to rationalize writing an entire paper to make that point.

[Figure]

The authors also claim in the abstract that "These results suggest that changes in model treatment of mixed-phase cloud processes and changes to Antarctic sea ice representation are likely causes of the shift towards larger ECS." But this is not proven anywhere in the paper — in fact, it's only mentioned twice and labeled as "speculation".

Line 73: The authors subtract the time series of the PI control run from the abrupt 4xCO2 run to account for drift of the underlying model run. This assumes that the modes of unforced variability are the same for 150 years in both runs, but I'm wondering how reasonable this is. Couldn't the huge RF from 4xCO2 change the internal modes of variability after 100 years of warming? I think it would be useful for the authors to comment on whether the results would be any different if the authors just subtracted the time-mean of the piControl run.

Line 93: I do not think it is correct to say that natural climate variability is a source of observational uncertainty. That needs to be rephrased.

Line 105: how is forcing calculated from the Gregory method? Is it just the Y intercept? Or are they making any other adjustments for fast responses?

I'm somewhat confused by section 3.2. Estimating the significance of differences between the means of two populations is a standard statistical problem. i.e, you can use some variant of a t-test to do that. I would think that this calculation should be described in one or two sentences. I do not understand why the authors have addresses this question with the approach in this section. There may be a reason I'm missing — if so, they should detail it.

Related to the previous comment, I don't understand why the median being larger than the mean matters. In their calculation, they reject ECS values > 10,000 K. What if they rejected values of ECS > 10 K? 10 K is only slightly less improbable, and I think choosing this lower threshold would reduce the difference between mean and median.

I don't doubt the result that the difference is significant, but I do think this section needs

to be rethought and perhaps reduced to a simple t-test.

Section 4.2: In this section, they compare zonal average feedbacks between the CMIP5 and CMIP6 ensembles. They need to add to the section a determination of where (at which latitudes) the differences between the ensemble averages are statistically significant. Then they can modify the discussion accordingly. For example, I'm not sure any of the differences in Fig. 6 are significant.

They should probably add a reference to Zelinka et al.'s new paper on the difference between the CMIP5 and 6 ensembles (https://agupubs.onlinelibrary.wiley.com/doi/full/10.1029/2019GL085782). It would be good to put the results of this paper into context with those results.

---

## Referee Comment (RC2) · Anonymous Referee #2 · 21 Feb 2020

Review for Flynn & Mauritsen

This manuscript analyzes differences in the climate sensitivity and transient climate response (TCR) between the CMIP5 and CMIP6 models. After showing that the increases in climate sensitivity and TCR of the CMIP6 models relative to the CMIP5 models is statistically significant, the authors aim to explain the underlying reasons for the increases. They conclude that the increase in climate sensitivity is likely related to changes in mixed-phase clouds and they suggest that TCR increased due to exaggerated warming after the 1970s.

[Figure]

In my opinion, the manuscript addresses an important and interesting topic, however, after Section 3 (which addresses the possibility that the increases in climate sensitivity and TCR may be due to chance), I find that the manuscript is only scratching the surface of several complex topics, though I do feel that it is going in the right direction and it points out several important issues. I also find that the text is not very detailed, and there are too many figures that don't directly address the "why"'s to a satisfactory degree in my opinion. Overall, I think the manuscript could benefit from a clearer "punchline" that is backed by solid analysis. Furthermore, the results for the analysis of climate sensitivity related to cloud feedbacks has already been noted in more detail by a recent publication by Zelinka et al. (2020) (not cited in the references), that looked at a similar number of models. I would recommend that the authors look for a clearer "punch line" for this paper, and to do a more in-depth analysis on it before this paper can be accepted for publication. Perhaps they could focus on and expand the analysis of the increase of TCR, or on the clear-sky feedbacks, for example.

Specific comments:

- I realize that Zelinka et al. (2020) was first published online on Jan. 3, 2020, but I think it's important for the authors to differentiate their work from this paper now that it has been published. First, how do the authors reconcile the fact that Zelinka et al. (2020) actually find that the increase in ECS in the CMIP6 models is statistically *insignificant*? Second, Zelinka et al. (2020) went further and performed a cloud feedback analysis of the CMIP5 and CMIP6 models. They found that besides the cloud optical depth feedback, the cloud amount feedback also played a large role in the increase in climate sensitivity. The authors "speculate" the possibility of cloud optical depth playing a central role in the increased climate sensitivity via mixed-phase cloud processes, but apparently Zelinka et al. (2020) had shown that cloud fraction changes play just as strong a role.

- What I'm left wondering is *why* aerosol cooling is stronger for the pre-1970 period but compensated for with greater post-1970 warming in CMIP6?

- Section 2.1: Why was an attempt to account for annual fluctuations applied to piControl but not for abrupt4xCO2?

---

## Referee Comment (RC3) · Anonymous Referee #3 · 21 Feb 2020

This manuscript presents an overview of ECS and and historical warming in a set of CMIP5 and CMIP6 models. The manuscript is well written, clear and concise. It describes some interesting findings. I would recommend publication after minor improvements as suggested below.

* There is significant overlap with the recently published paper Zelinka et al. (2020; doi:10.1029/2019GL085782). Given the close timing, this is not a serious problem. However, this manuscript should compare and contrast their findings to the ones in Zelinka. Ideally, the sets of CMIP5/CMIP6 models in this work should be a superset of the sets in Zelinka. I would also recommend listing models in Tables 1 and 2 with

[Figure]

the same alphabetical convention for easier comparison with their Tables S1 and S2. I also noticed that while most ECS values are close, some differ more substantially: EC-Earth3-Veg (4.33 vs 3.93) and SAM0-UNICON (3.72 vs 3.30). Is it perhaps because these models drift more than others, and thus the details of the drift correction matter more?

\* Line 72: SW and LW feedback: I assume the procedure applies to both all-sky and clear-sky feedback parameters discussed later? Is the drift correction the same as well?

\* Lines 80-81: There are additional significant volcanic eruptions (Santa Maria 1902; El Chichon, 1982) that fall within the period. Are they not excluded because they don't fall during the beginning or end portions of the periods over which averaging is performed? Please clarify.

\* Lines 85-86: the logic for the varying averaging length periods is not very clear. Please explain the reasoning behind these particular choices. Also, are the results sensitive to these choices?

\* Section 3.2 (lines 116-130). I had to read this section several times to really understand it. It could benefit from being rewritten more clearly. Some specific points:

- Lines 119-120: specify the mean of the Gaussian distribution for the feedback parameter.

- Lines 120-121: why 3.7 W/m2 with 10% standard deviation? The standard deviation for F_2x in Tables 1 and 2 is larger than 10% and the mean lower than 3.7 W/m2.

- Consider possibly swapping x-y axes in top panel of Figure 3 so that the black and red vertical lines align across the two panels.

\* Line 222: reference is missing year.

\* Figure 4: it's difficult to differentiate between black and dark gray lines. Why not use

[Figure]

the same color convention as in Figures 1 to 3: black for CMIP5 and red for CMIP6?

* Figure 7 caption or corresponding text: please clarify precisely what is being plotted.

* Figure 8 caption or corresponding text: "anomaly" with respect to what period?

* Figures 10 and 11: consider changing the figure aspect ratio to provide more resolution along the horizontal axes.

---

## Author Comment (AC1) · 16 May 2020

We would like to thank the reviewers for careful and thorough reading of this manuscript and for the thoughtful comments and constructive suggestions, which help to improve the quality of this manuscript. We provide point-by-point responses to each reviewer comment.

**Response to Reviewer 1**

**Reviewer Point P 1** — *This paper is an analysis of the differences between CMIP5 and CMIP6. It has relatively low information content, but I don't see anything technically wrong with it. My comments are therefore relatively minor.*

**Reviewer Point P 2** — *Overall, having read the paper, it was hard to determine what the top-line scientific message of the paper was. This problem is epitomized in the abstract. Per that, the most important conclusion is that the difference in ensemble averages between CMIP5 and CMIP6 could not have arisen by chance. Is that really the most important point in the paper? I do think it's a reasonable point to make, but it's a simple means test and it's hard to rationalize writing an entire paper to make that point.*

**Reply**: We believe there is a broad interest in understanding what might have caused the upward shift in ECS between CMIP5 and CMIP6, and even though our paper is being cautious concerning the conclusions that can be drawn at this stage we believe such an overview adds value to the literature. By choosing a broader scope for our paper, we have been able to highlight the effects of the increase in ECS between CMIP5 and CMIP6, including the different historical warming evolution suggesting changes to aerosol forcing between the two ensembles. This point seems to have been lost in the abstract, so we have modified it to make this surprising change in historical warming evolution more apparent for readers, and expanded the analysis in the corresponding section as much as reasonably possible. We have also expanded our analysis of clear-sky feedbacks to provide a firmer linkage among the zonal clear-sky shortwave feedbacks, sea ice coverage, and ECS to provide a firmer message to our paper.

**Reviewer Point P 3** — *The authors also claim in the abstract that "These results suggest that changes in model treatment of mixed-phase cloud processes and changes to Antarctic sea ice representation are likely causes of the shift towards larger ECS." But this is not proven anywhere in the paper — in fact, it's only mentioned twice and labeled as "speculation".*

**Reply**: Indeed, it is not possible to prove the root cause of these changes without detailed dissection of the model codes. Such an investigation is well beyond the scope of this paper as this would take years to do, and hence it is appropriate to label the finding as speculative at this stage, keeping in mind that it is a testable hypothesis. Future work should test these hypotheses and provide more detailed analysis of the causes.

**Reviewer Point P 4** — *Line 73: The authors subtract the time series of the PI control run from the abrupt 4xCO2 run to account for drift of the underlying model run. This assumes that the modes of unforced variability are the same for 150 years in both runs, but I'm wondering how reasonable this is. Couldn't the huge RF from 4xCO2 change the internal modes of variability after 100 years of warming? I think it would be useful for the authors to comment on whether the results would be any different if the authors just subtracted the time-mean of the piControl run.*

**Reply**: The rationale of removing the drift as it is estimated from the piControl simulations is that some modeling centers do not run their models to stationarity. This remaining drift is presumably of similar magnitude in both simulations. The reviewer is correct that an apparent drift which is really due to internal variability is not necessarily going to be the same, and doing so will introduce random errors that eventually average to zero. We have tested computing the ECS and feedback parameters subtracting the time-mean of piControl, rather than a linear regression over the piControl simulation, and it made no substantial difference. We agree it would be useful for readers to know this, and have added a sentence to Section 2.1.

**Reviewer Point P 5** — *Line 93: I do not think it is correct to say that natural climate variability is a source of observational uncertainty. That needs to be rephrased.*

**Reply**: We appreciate the reviewer pointing this out, and we have clarified in the text that the estimate of natural climate variability is in addition to rather than part of the observational uncertainty.

**Reviewer Point P 6** — *Line 105: How is forcing calculated from the Gregory method? Is it just the Y intercept? Or are they making any other adjustments for fast responses?*

**Reply**: We calculated forcing as one-half of the y-intercept of the regression over the 150 years of the abrupt4xCO2 simulation, following the Gregory method. This method does include what is often referred to as fast adjustments, insofar as they happen in much less than a year. The thus estimated forcing is, however, slightly low biased due to curvature of imbalance versus temperature found in several models.

**Reviewer Point P 7** — *I'm somewhat confused by section 3.2. Estimating the significance of differences between the means of two populations is a standard statistical problem. i.e, you can use some variant of a t-test to do that. I would think that this calculation should be described in one or two sentences. I do not understand why the authors have addressed this question with the approach in this section. There may be a reason I'm missing — if so, they should detail it.*

**Reply**: We understand how this section may be confusingly written, and appreciate the reviewer pointing this out. We have added more detail that should clarify our motivation and approach for comparing the CMIP ensemble means in the way that we have. In essence, because the underlying ECS distribution is rather positively skewed, we did not use a t-test or similar statistical test because they rely on the sample having an underlying normal distribution (and ECS does not). Instead, we assumed normality in the feedback and forcing distributions and used random sampling of the computed ECS distribution to assess the probability of attaining the CMIP5 or CMIP6 mean ECS by chance. This at least avoids the issue of the non-normality of the ECS distribution. This is how we obtain the very low probability of the CMIP5 and CMIP6 means being statistically the same.

**Reviewer Point P 8** — *Related to the previous comment, I don't understand why the median being larger than the mean matters. In their calculation, they reject ECS values >10,000 K. What if they rejected values of ECS >10 K? 10 K is only slightly less improbable, and I think choosing this lower threshold would reduce the difference between mean and median.*

**Reply**: The difference between median and mean ECS was critical to how we used the ECS distribution to compute the probability of obtaining the CMIP ensemble means, again due to the skewness of the

distribution, and was at first tricky to distinguish when we performed the analysis. This skew causes the median and mean to not necessarily be equal, as they would be in a Gaussian distribution; the ECS distribution computed with Equation 1 actually produced only the median rather than the mean ECS as the centroid of the distribution. We needed to be sure we were randomly sampling the correct parameter (we initially thought we were sampling the mean but later realized we were actually sampling the median ECS) in order to ultimately obtain the probability of the CMIP ensemble means. Then we could determine which median values corresponded to the CMIP ensembles mean, and thus find the probabilities. We tested rejecting values of ECS >10 K, and it did not reduce the difference between mean and median at the ECS values of interest to this analysis. The cut-off ECS value (10 K vs. 10,000 K) does begin to affect the difference between median and mean for ECS values above approximately 6 K, which are larger than all CMIP ECS values and rather unlikely. We see no reason to lower the threshold.

**Reviewer Point P 9** — *I don't doubt the result that the difference is significant, but I do think this section needs to be rethought and perhaps reduced to a simple t-test.*

**Reply**: We checked our statistical significance result against that from a t-test, and the means were still statistically different for the subsets of models included in the CMIP5 and CMIP6 ensembles examined in this work. It should be noted that such a test may not be appropriate for ECS due the reasons mentioned above, however.

**Reviewer Point P 10** — *Section 4.2: In this section, they compare zonal average feedbacks between the CMIP5 and CMIP6 ensembles. They need to add to the section a determination of where (at which latitudes) the differences between the ensemble averages are statistically significant. Then they can modify the discussion accordingly. For example, I'm not sure any of the differences in Fig. 6 are significant.*

**Reply**: We have determined where the ensemble averages are statistically significantly different, as the reviewer suggested, and visualized this in Figs. 5 and 6 as dashed lines on the CMIP5 and CMIP6 curves in the LW and SW panels (top and middle). The bottom panel was left as-is. Several regions in both the all-sky and clear-sky LW and SW zonal feedbacks were found to be statistically different between ensembles. The discussion has been modified to reflect this.

**Reviewer Point P 11** — *They should probably add a reference to Zelinka et al.'s new paper on the difference between the CMIP5 and 6 (https://agupubs.onlinelibrary.wiley.com/doi/full/10.1029/ 2019GL085782). It would be good to put the results of this paper into context with those results.*

**Reply**: We agree, since Zelinka et al. (2020) was published at a similar time as our initial submission; discussion and comparison to their results have been added to the end of our discussion of the global and zonal feedbacks (in Section 4.2), as our results and theirs are complementary.

**Response to Reviewer 2**

**Reviewer Point P 1** — *This manuscript analyzes differences in the climate sensitivity and transient climate response (TCR) between the CMIP5 and CMIP6 models. After showing that the*

*increases in climate sensitivity and TCR of the CMIP6 models relative to the CMIP5 models is statistically significant, the authors aim to explain the underlying reasons for the increases. They conclude that the increase in climate sensitivity is likely related to changes in mixed-phase clouds and they suggest that TCR increased due to exaggerated warming after the 1970s.*

**Reviewer Point P 2** — *In my opinion, the manuscript addresses an important and interesting topic, however, after Section 3 (which addresses the possibility that the increases in climate sensitivity and TCR may be due to chance), I find that the manuscript is only scratching the surface of several complex topics, though I do feel that it is going in the right direction and it points out several important issues. I also find that the text is not very detailed, and there are too many figures that don't directly address the "why"'s to a satisfactory degree in my opinion. Overall, I think the manuscript could benefit from a clearer "punchline" that is backed by solid analysis. Furthermore, the results for the analysis of climate sensitivity related to cloud feedbacks has already been noted in more detail by a recent publication by Zelinka et al. (2020) (not cited in the references), that looked at a similar number of models. I would recommend that the authors look for a clearer "punch line" for this paper, and to do a more in-depth analysis on it before this paper can be accepted for publication. Perhaps they could focus on and expand the analysis of the increase of TCR, or on the clear-sky feedbacks, for example.*

**Reply**: We sought a broader scope for our paper, covering the increase in ECS between CMIP6 and CMIP5, its causes and effects – mainly, the different historical warming evolution between the two ensembles that suggest changes to aerosol forcing – to point to top-line changes and areas requiring further investigation in future work. We agree that it is important to have focused and detailed analysis of particular processes that have changed, such as was done in Zelinka et al. (2020) with regards to the clouds feedbacks and how they have driven the increase in ECS, this is important for our understanding of model behavior just as having more of an overview of ensemble changes in several processes is also of value. That said, we understand the reviewer's suggestion to look for a clearer "punch line," and have expanded our analysis as much as we reasonably can; in particular, we have expanded our analysis of the clear-sky feedbacks to provide a firmer linkage among the zonal clear-sky shortwave feedbacks, sea ice coverage changes, and ECS.

Zelinka et al. (2020) has now been added to the references and discussion of their results included in the appropriate sections; the article was not previously included in our paper because it had not yet been published online as of the time of initial writing and submission of our paper.

**Reviewer Point P 3** — *I realize that Zelinka et al. (2020) was first published online on Jan. 3, 2020, but I think it's important for the authors to differentiate their work from this paper now that it has been published. First, how do the authors reconcile the fact that Zelinka et al. (2020) actually find that the increase in ECS in the CMIP6 models is statistically insignificant? Second, Zelinka et al. (2020) went further and performed a cloud feedback analysis of the CMIP5 and CMIP6 models. They found that besides the cloud optical depth feedback, the cloud amount feedback also played a large role in the increase in climate sensitivity. The authors "speculate" the possibility of cloud optical depth playing a central role in the increased climate sensitivity via mixed-phase cloud processes, but apparently Zelinka et al. (2020) had shown that cloud fraction changes play just as strong a role.*

**Reply**: We agree it is important to compare and contrast our results with Zelinka et al. (2020), since they were published soon after our initial submission. We have added discussion comparing our feedback

results to theirs at the end of Section 4.2, as our results are complementary, and they expanded upon cloud feedbacks (and we have added discussion detailing their finding of the importance of the cloud amount and optical depth albedos) while we examined clear-sky zonal feedbacks in more depth. Our work is complementary and overall in agreement, but different emphasis has been placed on different aspects of the changes between the CMIP5 and CMIP6 ensembles. For example, we did not delve into feedback decomposition, which has both advantages and disadvantages, but instead expanded upon the clear-sky feedbacks and examined ensemble changes in historical warming. A kernel-based feedback analysis can provide detailed information about cloud processes, but may also yield large residuals relative to the actual all-sky fluxes. Therefore, we consider an all-sky and clear-sky analysis complementary.

About the statistical significance of the increase in mean ECS, we took a different approach to assessing the statistical significance of the difference in the CMIP ensemble means from Zelinka et al., due to the inherent skewness of the underlying ECS distribution; please refer to our responses to Reviewer 1, Points 7 and 8, who raised similar points about our approach. We wanted to avoid any impact of assuming normality of the ECS distribution on the results of statistical significance tests such as the commonly used variants of t-tests, which can be affected by non-normality. In addition to this, while most of the models we examine are the same, we include some models that Zelinka et al. do not, and vice versa. This might also impact why Zelinka et al. did not find a statistically significant difference for the specific subset of models they examined, but that we did find a significant difference for our subset, both with our approach and with a t-test for difference in means; their non-significance therefore does not necessarily contradict our finding of a significant difference. We have included a brief explanation of this in Section 3.2.

**Reviewer Point P 4** — *What I'm left wondering is why aerosol cooling is stronger for the pre-1970 period but compensated for with greater post-1970 warming in CMIP6?*

**Reply**: Aerosol precursor and direct emissions stagnated after the 1970s, whereafter aerosol cooling only evolved slowly. During the same period of time greenhouse gas forcings continued to increase. Presumably a larger fraction of CMIP6 models now include more of the cloud-aerosol interactions (indirect effects) and so may have an overall larger aerosol-induced cooling magnitude, though the evolution in time is tied to the emissions evolution.

We agree it would be useful to know the reasons behind greater aerosol cooling in the pre-1970 period compared to post-1970, and why the compensation with greater warming occurs post-1970 in CMIP6 relative to CMIP5. We believe RFMIP simulations are the most appropriate to use to answer this question. Unfortunately, RFMIP simulations are not avaialable for CMIP5, so CMIP6 cannot be compared to CMIP5, and we therefore cannot adequately answer why this difference exists between CMIP5 and CMIP6. The best we can do here is to demonstrate that a stronger aerosol cooling might be the case, and leave it to future work to sort out the causes.

**Reviewer Point P 5** — *Section 2.1: Why was an attempt to account for annual fluctuations applied to piControl but not for abrupt4xCO2?*

**Reply**: Annual fluctuations as well as drift were removed from piControl instead of also abrupt4xCO2 because some models were not run to equilibrium in their piControl simulations. We assume the remaining fluctuations and drift are of similar magnitude in both simulations, and so do not remove them in abrupt4xCO2.

**Response to Reviewer 3**

**Reviewer Point P 1** — *This manuscript presents an overview of ECS and and historical warming in a set of CMIP5 and CMIP6 models. The manuscript is well written, clear and concise. It describes some interesting findings. I would recommend publication after minor improvements as suggested below..*

**Reviewer Point P 2** — *There is significant overlap with the recently published paper Zelinka et al. (2020; doi:10.1029/2019GL085782). Given the close timing, this is not a serious problem. However, this manuscript should compare and contrast their findings to the ones in Zelinka. Ideally, the sets of CMIP5/CMIP6 models in this work should be a superset of the sets in Zelinka. I would also recommend listing models in Tables 1 and 2 with the same alphabetical convention for easier comparison with their Tables S1 and S2. I also noticed that while most ECS values are close, some differ more substantially: EC- Earth3-Veg (4.33 vs 3.93) and SAM0-UNICON (3.72 vs 3.30). Is it perhaps because these models drift more than others, and thus the details of the drift correction matter more?*

**Reply**: Comparison to the results from Zelinka et al. (2020) have been added to our discussion of the global and zonal feedbacks (Section 4.2), as our results and theirs are complementary. However, it should be noted that the scopes of our paper and theirs are different – Zelinka et al. focused on the cloud feedbacks while we took a broader overview and examined all-sky and clear-sky feedbacks and historical warming within the ensembles.

  We also re-examined our ECS values, and found that some model output files were corrupted, and their ECS values have been corrected accordingly and are overall in better agreement with Zelinka et al. (this includes the two models the reviewer pointed out: EC-Earth3-Veg was corrected from an ECS of 3.93K to 4.17K, and SAM0-UNICON, corrected from 3.30 to 3.67). Other differences in ECS magnitude are caused by differences in methodology coupled with some model drift. Corrections for drift to the ECS values, however, did not impact the conclusions of this paper. We have also re-ordered Tables 1 and 2 to be in alphabetical order, as in Zelinka et al.

**Reviewer Point P 3** — *Line 72: SW and LW feedback: I assume the procedure applies to both all-sky and clear-sky feedback parameters discussed later? Is the drift correction the same as well?*

**Reply**: That is correct, the same procedures and drift correction were applied to compute the all-sky and clear-sky SW and LW feedbacks.

**Reviewer Point P 4** — *Lines 80-81: There are additional significant volcanic eruptions (Santa Maria 1902; El Chichon, 1982) that fall within the period. Are they not excluded because they don't fall during the beginning or end portions of the periods over which averaging is performed? Please clarify.*

**Reply**: We sought to exclude only those 20th century volcanic eruptions that penetrated the stratosphere, though not excluding Santa Maria and El Chichon was an oversight. We have now also excluded these two eruptions, and amended the figure and text.

**Reviewer Point P 5** — *Lines 85-86: the logic for the varying averaging length periods is not very clear. Please explain the reasoning behind these particular choices. Also, are the results sensitive to these choices?*

**Reply**: The differencing periods to compute the warming within the pre- and post-1970s periods were chosen to be roughly half each period, taking into account volcanic eruptions. Yes, the results are somewhat sensitive to the choices of averaging periods, but not sensitive enough to have a significant impact on the main results from this analysis, as the pre- and post-1970s warmings are each dominated by a different, domninant type of forcing (aerosol forcing for the pre-1970s period and GHG forcing for the post-1970s period).

**Reviewer Point P 6** — *Section 3.2 (lines 116-130). I had to read this section several times to really understand it. It could benefit from being rewritten more clearly.*

**Reply**: We appreciate the reviewer pointing out this problem and we have added more explanation for our motivation for the approach we used, as well as details about the procedure itself; please see our responses to Reviewers 1 and 2, who brought up similar points.

**Reviewer Point P 7** — *Some specific points:Lines 119-120: specify the mean of the Gaussian distribution for the feedback parameter.*

**Reply**: We have added the range of values for the mean feedback parameter we examined, as we tested many to have higher confidence in the median ECS values that corresponded with the CMIP5 and CMIP6 mean values. We have also included the mean feedback parameters that correspond specifically to those median ECS values.

**Reviewer Point P 8** — *Lines 120-121: why 3.7 W/m2 with 10% standard deviation? The standard deviation for $F_{2x}$ in Tables 1 and 2 is larger than 10% and the mean lower than 3.7 W/m2.*

**Reply**: We used these values based on process understanding (e.g. Etminan et al. 2016) because forcing was not a significant driver of the increase in ECS between CMIP5 and CMIP6, and so avoided complicating our approach to compute the statistical significance of the difference between the two CMIP means by using forcing estimates based on the CMIP ensembles.

**Reviewer Point P 9** — *Consider possibly swapping x-y axes in top panel of Figure 3 so that the black and red vertical lines align across the two panels.*

**Reply**: Though we understand why a reader may want to have alignment between the vertical lines in each panel, we think that is is somewhat easier to see how the mean departs from the median in the top panel in the current orientation.

**Reviewer Point P 10** — *Line 222: reference is missing year.*

**Reply**: The missing year has been added.

**Reviewer Point P 11** — *Figure 4: it's difficult to differentiate between black and dark gray lines. Why not use the same color convention as in Figures 1 to 3: black for CMIP5 and red for CMIP6?*

**Reply**: We have changed the line colors to follow the color convention of the previous plots to make it easier to differentiate between the two ensemble averages.

**Reviewer Point P 12** — *Figure 7 caption or corresponding text: please clarify precisely what is being plotted.*

**Reply**:  We have clarified the figure caption.

**Reviewer Point P 13** — *Figure 8 caption or corresponding text: "anomaly" with respect to what period?*

**Reply**:  We have clarified in the caption that this is the abrupt4xCO2 surface temperature anomaly with respect to piControl (computed in the same way as the annual-mean, global-mean surface temperature anomalies used to compute ECS, but for zonal- rather than global-mean temperature).

**Reviewer Point P 14** — *Figures 10 and 11: consider changing the figure aspect ratio to provide more resolution along the horizontal axes.*

**Reply**:  We have widened the aspect ratio to make the x-axis and the plot easier to read.